# A Porphyrin-Based Covalent Organic Framework as Metal-Free Visible-LED-Light Photocatalyst for One-Pot Tandem Benzyl Alcohol Oxidation/Knoevenagel Condensation

**DOI:** 10.3390/nano13030558

**Published:** 2023-01-30

**Authors:** Sara Oudi, Ali Reza Oveisi, Saba Daliran, Mostafa Khajeh, Amarajothi Dhakshinamoorthy, Hermenegildo García

**Affiliations:** 1Department of Chemistry, Faculty of Sciences, University of Zabol, Zabol P.O. Box 98615-538, Iran; 2Departamento de Quimica, Universitat Politècnica de València, Av. De los Naranjos s/n, 46022 Valencia, Spain; 3School of Chemistry, Madurai Kamaraj University, Madurai 625021, Tamil Nadu, India; 4Instituto Universitario de Tecnología Química, Consejo Superior de Investigaciones Científicas-Universitat Politecnica de Valencia, Av. De los Naranjos s/n, 46022 Valencia, Spain

**Keywords:** covalent organic framework, porous materials, porphyrin, boronic ester linkages, cascade reaction, carbon–carbon coupling reaction, photooxidation, multifunctional catalyst

## Abstract

A porphyrin-based covalent organic framework (COF), namely Porph-UOZ-COF (UOZ stands for the University of Zabol), has been designed and prepared via the condensation reaction of 5,10,15,20-tetrakis-(3,4-dihydroxyphenyl)porphyrin (DHPP) with 1,4-benzenediboronic acid (DBBA), under the solvothermal condition. The solid was characterized by spectroscopic, microscopic, and powder X-ray diffraction techniques. The resultant multifunctional COF revealed an outstanding performance in catalyzing a one-pot tandem selective benzylic C-H photooxygenation/Knoevenagel condensation reaction in the absence of additives or metals under visible-LED-light irradiation. Notably, the catalytic activity of the COF was superior to individual organic counterparts and the COF was both stable and reusable for four consecutive runs. The present approach illustrates the potential of COFs as promising metal-free (photo) catalysts for the development of tandem reactions.

## 1. Introduction

Nature is a source of inspiration for new ideas. Over the years, researchers have been trying to mimic nature’s generative models and their high potential applications. Porphyrins, such as cytochrome P450 and chlorophylls, are universally used in nature as co-factor/active centers for oxidation and photosynthetic processes [1,2,3]. However, homogenous, synthetic porphyrin-based catalysts irreversibly self-deactivate or self-destruct in solutions by forming inactive bimolecular species [4,5]. In order to address this problem, porphyrins have been tethered on solid supports or incorporated in porous architectures for their use in heterogeneous catalysis [6,7,8], water remediation [9], and hydrogen production [10]. Nevertheless, in spite of these successful achievements, some of the materials reported display a low active site density per mass unit, which is detrimental to achieving high reaction rates. Another main challenge in catalysis is the merging of the advantages of both homogeneous and heterogeneous catalytic processes to preserve, or enhance, the activity and selectivity of the molecular catalysts while allowing its recovery, reuse, and easy product separation after reaction [11,12,13,14,15]. In this regard, the construction of stable porous solids by assembling a high concentration of catalytically active porphyrin building units is still challenging and highly demanding. 

Benefiting from covalent linking of organic building blocks, covalent organic frameworks (COFs) have emerged as a new class of soft, porous materials, with applications in catalysis, sensing, adsorption, gas/energy storage, and other various fields [14,16,17,18,19]. Similar to those documented in metal organic frameworks (MOFs), COFs represent attractive properties such as crystallinity, a large surface area, a high porosity, structural versatility, low density, and a high degree of structural stability [14,18,20]. A common strategy for constructing COFs is to form boron–oxygen joints between two complementary organic building blocks through the dynamic covalent chemistry [21,22,23]. As a result of the convenient synthesis, the construction of boron-containing COFs has grown rapidly, by employing the organic linkers and the synthetic conditions [24].

Photocatalysis is a green and sustainable technology that is becoming extensively researched due to the advantages of using natural sunlight as an energy source, the simple infrastructure required, the low-energy consumption, and the mild reaction conditions. The application of light-assisted organic synthesis has also been growing at a fast pace over the past few years [16,25,26]. On the other hand, tandem reactions are a current front in chemical engineering, since it avoids unnecessary process steps, simplifies the reaction workup, and represents the intensification of a multi-step process [27,28,29]. Tandem reactions usually require multifunctional catalysts that are able to promote more than one single reaction [30]. The formation of a carbon–carbon bond through the selective aerobic oxidation of alcohols and consecutive Knoevenagel condensation reaction is one of the most fundamental transformations in organic chemistry [31,32,33,34]. In this context, various heterogeneous catalysts have been reported but possess numerous shortcomings, including poor stability, precious metals, and/or metal contamination, thereby hampering their practical applications [7,8,35,36]. Thus, there is a great demand for developing greener, more efficient, and more sustainable heterogeneous multifunctional catalytic systems.

In this contribution, we designed and prepared a new porphyrin-based covalent organic framework (Porph-UOZ-COF; UOZ: University of Zabol), which was employed as a metal-free heterogeneous visible-LED-light-active photocatalyst for tandem aerobic alcohol oxidation–Knoevenagel condensation reaction under mild conditions in the absence of any added chemicals or co-catalyst. The COF-based poly(boronate ester) (Porph-UOZ-COF) was synthesized by the condensation of DHPP and DBBA under solvothermal conditions. From a structural point of view, the main factor in the framework is the presence of precisely spaced porphyrins (as dual redox active and basic sites) and boronic esters (as Lewis acidic sites) in a high density. Porph-UOZ-COF exhibits superior catalysis activity in comparison with the homogeneous COF precursors, which can be mostly ascribed to the synergistic effect of immobilized catalytically active sites within the framework. Furthermore, the ordered porous structure of Porph-UOZ-COF with interconnected micropores and mesopores makes favorable the mass diffusion/transfer processes, thereby further promoting the improvement of catalytic activity.

## 2. Experimental Section

### 2.1. Materials

3,4-Dimethoxy benzaldehyde, propionic acid, pyrrole, boron tribromide, 1,4-phenylenediboronic acid (DBBA), methanol, ethanol, dichloromethane, ethyl acetate, dioxane, mesitylene, benzyl alcohols, and malononitrile were purchased from Sigma-Aldrich, Merck, and were used without further purification. 5,10,15,20-Tetrakis-(3,4-dihydroxyphenyl)porphyrin (DHPP) was synthesized by a two-step process according to the procedure reported in the literature [7,37].

### 2.2. Characterization Techniques

Powder X-ray diffraction (PXRD) spectra were recorded on a Philips PANalytical X’Pert XRD diffractometer (Almelo, Netherlands). The surface morphology was determined by field emission scanning electron microscopy (FESEM, TESCAN MIRA3, Czech Republic). Brunauer–Emmett–Teller (BET) specific surface area and pore size distributions (PSDs) of the solid were measured by isothermal nitrogen adsorption–desorption method at −196 °C using an ASAP 2020 instrument (Micromeritics Instrument Corporation, Norcross, GA, USA). Thermal stability was measured using a Mettler Toledo TGA/DSC instrument (Giessen, Germany) under N_2_ gas. Electron spin resonance (EPR) spectra were recorded with a Bruker instrument. X-ray photoelectron spectra (XPS) were obtained on a SPECS spectrometer coupled with a Phoibos 150 9MCD as detector using a non-monochromatic X-ray source (Al). Diffuse reflectance UV-Vis absorption spectrum (DRS) was collected in a Shimadzu UV-Vis spectrophotometer using BaSO_4_ as a reference. The visible light irradiation was performed using LED lamps (280 power, 3.2 V, 1 W, λ > 420 nm) in a tube-shaped vessel, 32,000 LUX. The solid-state ^11^B NMR spectrum was recorded with a Bruker Avance III HD 400 WB spectrometer using 90° pulses, a recycle delay of 1 s, and a spinning rate of 20 kHz, while solution NMR spectra were recorded on a Bruker Avance DPX-250 NMR. Varian gas chromatography/mass spectrometry (GC/MS) is used to detect benzaldehyde as intermediate.

### 2.3. Synthesis of Covalent Organic Framework (Porph-UOZ-COF)

DHPP (7 mg, 0.62 mmol) and DBBA (15 mg, 0.04 mmol) were poured into a 10 mL Pyrex vial containing a mixture of 1,4-dioxane and mesitylene (2 mL, 9:1 in volume). Then, the vial was degassed under vacuum and placed in an oil bath at 120 °C for 5 days. The resulting solid was collected by centrifugation and was washed several times with acetone before being dried under vacuum at 150 °C.

### 2.4. One-Pot Tandem Photocatalytic Carbon–Carbon Forming Reaction of Alcohols and Malononitrile

Benzyl alcohol (1 mmol), malononitrile (1.3 mmol), and Porph-UOZ-COF (20 mg) were mixed in CH_3_CN:H_2_O (4:1, 3 mL) under a home-made visible LED irradiation system allowing stirring of the reaction vial. An oxygen balloon was connected to the vial to supply excess O_2_ during the reaction. The reaction progress was monitored by TLC (elution: n-hexane/ethyl acetate). When the reaction completed, the resulting mixture was centrifuged to separate the catalyst before the solvent evaporation. Then, the crude product was recrystallized from ethanol. In the reusability tests, Porph-UOZ-COF was washed with acetonitrile/ethyl acetate and dried under reduced pressure at 100 °C for 4 h. Detailed characterization data for all the compounds are given in the Appendix A.

## 3. Results and Discussion

### 3.1. Characterization of Porph-UOZ-COF

A free-base porphyrin COF (Porph-UOZ-COF) was assembled via a condensation reaction between DHPP (monomer 1) and DBBA (monomer 2) as shown in Figure 1.

The construction of boronate ester bonds (C_2_O_2_B ring) in the COF was established by an FT-IR spectroscopy by recording the characteristic peaks corresponding to the vibrational C-O bonds at 1115 cm^−1^, and the strong peak at 1344 cm^−1^ corresponding to the B-O stretching within the C_2_O_2_B ring (Appendix A). The morphology of the COF particles was studied by a scanning electron microscopy (SEM). The SEM images present irregular round-shaped aggregates with an average size of 20–100 nm (Figure 1).

Porosity properties were determined by nitrogen isotherms and BET surface areas. The gas adsorption measurements of the COF (Figure 2) indicate the presence of mesopores in addition to the micropores. The BET surface area of the solid was estimated as 130 m^2^/g (Langmuir surface area = 203 m^2^/g). In addition, the density functional theory (DFT) calculations showed that the pore size distribution ranges from micropores to mesopores (1.8–2.4 nm, sharply at 1.2 and 1.5 nm, and above 45 nm). The porosity allows the reactant molecules to easily reach the catalytically active sites inside the material.

XRD pattern of the prepared polymer (Figure 3, bottom) revealed diffraction signals at about 7.5 and 21°, which indicated a staggered model (AB stacking) similar to the AB simulated stacking of the ZnP-COF as a porphyrin-based COF in a related literature report [22].

The thermogravimetric analysis (TGA) was used to evaluate the thermal stability of the Porph-UOZ-COF catalyst in a nitrogen atmosphere. As shown in Appendix A, Porph-UOZ-COF decomposes at 500 °C with a total weight loss of ~34%. Additionally, the TGA profile shows that there is a slight weight loss of ~10% caused by the removal of H_2_O/solvent at the temperature ranging from 25 °C to 210 °C. The main weight loss of approximately 20%, measured at temperatures between 300–500 °C, corresponds to the total decomposition of the organic material followed by the breaking down of the framework.

XPS is a quantitative surface analysis technique, which can be used to determine elemental composition and the environment of the elements existing on the sample’s surface. Figure 4 shows the C 1s, N 1s, O 1s, and B 1s regions of the Porph-UOZ-COF’s XPS spectra. The C 1s peak (84.1 wt %) displays three carbon families at 284.6, 285.2, and 288.2 eV, attributable to sp^2^ C (-C=C-), sp^3^ C (-C-C-)/-C-N-, and -C=O functionalities, respectively [7,38]. Figure 4 also shows that the N 1s spectrum (2.4 wt %) can be fitted with binding energies of nitrogen atoms of pyrroles of the porphyrin monomers (mainly at 400.1 eV) [39]. The O 1s spectrum (12.4 wt %) of Porph-UOZ-COF exhibits two obvious components at around 532.1 eV and 533.2 eV for -C-O species and -C=O groups [7,38]. Notably, the XPS spectrum of the B 1s binding energy (1.1 wt %) revealed a peak at 191.7 eV [40], which agrees with the expected value for B atoms in the pentagonal C2O2B ring.

To complete the Porph-UOZ-COF characterization, the solid-state ^11^B NMR spectrum [41,42] of the sample was also recorded (Figure 5). As shown in Figure 5, the COF shows two distinct signals that arise probably from either the C_2_O_2_B ring (mainly) or the B_3_O_3_ ring [43,44]. The presence of B peaks in the XPS and solid-state ^11^B NMR indicates the formation of O-B bonds caused by the condensation of DHPP with DBBA, confirming further COF synthesis.

### 3.2. Porph-UOZ-COF-Catalyzed One-Pot Tandem Carbon–Carbon Coupling Reaction

After successful preparations of porous COF, we decided to investigate its potential activity for the tandem benzylic alcohol oxidation/Knoevenagel reaction, starting from alcohols and malononitrile, under visible-LED-light irradiation. To optimize the reaction conditions, benzyl alcohol and malononitrile were selected as substrates and Porph-UOZ-COF as the multifunctional (oxidation and condensation) catalyst. The results are presented in Table 1. The model reaction was first attempted using the prepared COF as the photocatalyst, the molecular oxygen as the green oxidant (one atmosphere pressure), and the acetonitrile as the solvent. The reaction affords a high yield of the target 2-benzylidenemalononitrile (named α,β-unsaturated nitrile) with a high selectivity and high conversion of benzyl alcohol (>99%); however, it requires a long reaction time (Table 1, entry 1). In addition, an analysis of the reaction mixture showed the detection of benzaldehyde as an intermediate (see Appendix A). To further improve the reaction conditions, the reaction was optimized at different solvents (entries 1–4). These optimization studies showed that the best reaction medium was a mixture of CH_3_CN and H_2_O in which the reaction preceded smoothly, giving the expected product an excellent yield within a shorter reaction time (Table 1, entry 2). Dehydration reactions have been reported in water [45,46,47] with notable efficiency using hydrophobic catalysts. There are several merits in such catalysis: (1) providing hydrophobic pockets, such as enzymes, (2) condensing organic substrate therein, (3) providing intermediate stabilization, and (4) providing weak product binding. We thus assumed that that hydrophobic COF framework could fulfill most of these features. On the other hand, the use of an aqueous solution as the reaction solvent makes the process an environmentally friendly chemical process.

The tandem reaction between benzyl alcohol and malononitrile was checked in absence of a catalyst or light as control reactions, whereby no product was observed (Table 1, entries 5 and 6). Notably, in the additional controls that used the COF monomers and OMe derivative of DHPP as catalysts instead of the active Porph-UOZ-COF, the product yield was low, or no progress occurred under these conditions (entries 7–9). These control experiments suggest that porphyrin groups in Porph-UOZ-COF act as photocatalytically active centers for this reaction. In addition, the higher product yield promoted by Porph-UOZ-COF than that of the corresponding homogeneous catalysts DHPP (or OMe derivative of DHPP) is attributed to the prevention of the well-known, efficiency diminishing, aggregation capability of porphyrin groups by the construction of the COF. Furthermore, it can also relate to the site isolation of porphyrin units in the framework. The reaction was found to be very slow and was afforded a lower product yield with a lower LED intensity (entry 10).

The scope of Porph-UOZ-COF to promote the tandem alcohol oxidation–Knoevenagel condensation was then investigated using various benzyl alcohols substituted with Me, Cl, and NO_2_ groups. As shown in Table 2, these tandem coupling reactions produced a high yield of the target α,β-unsaturated nitrile in reaction times of 60–70 h. The nature of functional groups showed no significant influence on the final product yield and reaction time (Table 2, entries 1–4).

Additionally, the advantages of porous framework were compared with those of reported catalysts for the cascade oxidation–Knoevenagel condensation reaction (Table 3). 

These comparisons clearly indicate that the present catalyst promotes this tandem reaction without the use of any transition metal and that the reaction medium involves water as a solvent. Furthermore, the Porph-UOZ-COF has noticeable merits for the synthesis of 2-benzylidenemalononitriles, as the reaction is carried out at room temperature, does not require high catalyst loading, neither additives, nor precious/toxic metals, and the reaction work-up is not tedious. Thus, Porph-UOZ-COF performs similarly to an analogous Fe porphyrin MOF (91% α,β-unsaturated nitrile yield in 24 h) [7], but the advantage with Porph-UOZ-COF is that it does not have any transition metal and is based on a COF structure rather than metal-coordination bonds that renders the material stable in a wider range of solvents, such as water. Compared to the most efficient CsCu_2_I_3_@PCN-222(Fe) composite (96% α,β-unsaturated nitrile yield in 15 h) [8], the present Porph-UOZ-COF catalyst enjoys a less elaborative synthesis than tedious procedure to incorporate the mixed halide perovskites inside the MOF while exhibiting similar performance. In addition, it should be noted that Porph-UOZ-COF is a recyclable catalyst. Furthermore, visible light energy is used as a safe, residue-less, and clean energy that can derive from renewable sources. 

The recyclability of Porph-UOZ-COF was assessed for the tandem photooxidation–Knoevenagel coupling reaction between benzyl alcohol and malononitrile. Porph-UOZ-COF was easily recovered by centrifugation, washed, and dried under a vacuum at 100 °C to be ready for the next run. The catalyst could be reused four times, and the product yield decreased gradually from 93 to 87% (Figure 6). This decrease is most probably due to incomplete catalyst recovery and catalyst mass loss rather than partial deactivation by pore blocking. 

The reused solid was characterized by an FT-IR spectroscopy (Appendix A), SEM (Figure 1), and PXRD (Figure 3; top one). All these data confirm that the structure is maintained during the catalytic cycles (Appendix A). These data indicate that Porph-UOZ-COF is a stable and active catalyst for the tandem reaction.

A probable reaction mechanism for the photocatalytic formation of 2-benzylidenemalononitriles promoted by Porph-UOZ-COF is proposed in Figure 7a. When the COF is exposed to visible light, it is excited to a higher energy state and produces an excited COF (COF*) consisting of localized triplet excitons at the porphyrin chromophore. Then, singlet oxygen (^1^O_2_) [15] as a reactive oxygen species would be generated from the energy transfer (ET) of triplet COF* excitons to molecular oxygen. The resulting ^1^O_2_ oxidizes selectively benzyl alcohol to benzaldehyde. Subsequently, boron Lewis acid [24,56] and Lewis base free-base porphyrin [57] within the framework, behaving as a bifunctional catalyst, will activate the aldehyde carbonyl group and the malononitrile methylene group, respectively, to react together and produce the desired product after removal of a water molecule. 

To gain deeper insights into the possible reaction mechanism, we have performed quenching tests [58] with different scavengers, including sodium azide, 1,4-benzoquinone, and isopropanol, to understand the influence of the reactive oxygenated species that dominate the photocatalytic process. The addition of sodium azide to the reaction exhibited significant inhibition of the yield, suggesting the involvement of singlet oxygen (^1^O_2_) as an active species for this system (Table 4, entry 1). On the other hand, the reaction was not entirely inhibited by sodium azide, which indicates that some other oxidation pathways other than ^1^O_2_ might also be involved in this reaction. The use of 1,4-benzoquinone and isopropanol as scavengers decreased slightly the product yield (Table 4, entries 2 and 3), demonstrating that superoxide radical anion (O_2_^•−^) and ^•^OH were also involved, but they are not dominant reactive species in the reaction. These quenching experiments confirm that ^1^O_2_ plays a key role in the photocatalytic system.

The generation of singlet oxygen (^1^O_2_) [15,59,60] during the reaction was also confirmed by an electron paramagnetic resonance (EPR) spectroscopy using a selective ^1^O_2_ spin trap. When 2,2,6,6-tetramethylpiperidine (TEMP) [8,61] was used as the specific scavenger for ^1^O_2_ under the conditions of the photocatalytic reaction, a characteristic three-line signal corresponding to the generation of persistent TEMPO radical (a strong 1:1:1 triplet signal) derived from the ^1^O_2_ oxidation of TEMP was observed (Figure 7b), confirming the photocatalytic activity of Porph-UOZ-COF under aerated conditions to form ^1^O_2_ and the suggested mechanism.

## 4. Conclusions

The present study has shown the synthesis of a porous porphyrin-based COF (Poph-UOZ-COF) by direct condensation of DBBA with DHPP. Further, Porph-UOZ-COF showed the catalytic activity for a one-pot tandem photocatalytic C-C coupling condensation reaction because of the efficient synergy between the production of reactive singlet oxygen and bifunctional acid-base sites. This tandem coupling operates at room temperature, and it does not require high catalyst loading, additives, or any metal compared to other reported polymers. In addition, Porph-UOZ-COF could be recycled without obvious changes to the structure and with a minor decrease in activity while using safe and clean visible light. This work illustrates the potential for the rational design of multifunctional COFs for their application in tandem reactions.

## Data Availability

Not applicable.

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
