# Peer review of "A Porphyrin-Based Covalent Organic Framework as Metal-Free Visible-LED-Light Photocatalyst for One-Pot Tandem Benzyl Alcohol Oxidation/Knoevenagel Condensation"

_nanomaterials, 2023, doi:10.3390/nano13030558_

Round 1
Reviewer 1 Report
The manuscript entitled “A porphyrin-based covalent organic framework for metal-free visible-LED-light driven one-pot tandem benzylic C-H oxygenation/Knoevenagel condensation reaction” by Oveisi and García et al described the synthesis and characterization of a porphyrin-based COF, namely Porph-UOZ-COF, which exhibited outstanding performance in catalyzing one-pot tandem selective benzylic C-H photooxygenation/Knoevenagel condensation reaction. Such photo-catalytic activity of a COF is encouraging. Probable mechanism of the tandem process of photocatalytic alcohol oxidation/Knoevenagel condensation promoted by Porph-UOZ-COF has been explored. I believe that this manuscript can be accepted for publication after some of the issues commented on below have been well addressed.
1. The abbreviations of the monomers, DHPP and DBBA, should be provided at the first time its full name appears. The monomer abbreviations should also be labeled in Scheme 1. The synthesis of DHPP in Scheme 1 is not given in detail in this manuscript and this part in Scheme 1 is suggested to be deleted.
2. In general, simulated X-ray powder diffraction pattern of the COF structure should be given for comparison, indicating that the obtained crystalline sample is the target product.
3. There are several types of reactive oxygen species, but only singlet oxygen is mentioned in the manuscript. Are there any other reactive oxygen species detected? The specific scavengers to reactive oxygen species should be added to the reaction process to observe the catalytic performance variation.
4. Some works on COFs for photocatalysis should be cited and discussed, e.g., Molecules, 2022, 27(22), 8002.
Author Response
The manuscript entitled “A porphyrin-based covalent organic framework for metal-free visible-LED-light driven one-pot tandem benzylic C-H oxygenation/Knoevenagel condensation reaction” by Oveisi and García et al described the synthesis and characterization of a porphyrin-based COF, namely Porph-UOZ-COF, which exhibited outstanding performance in catalyzing one-pot tandem selective benzylic C-H photooxygenation/Knoevenagel condensation reaction. Such photo-catalytic activity of a COF is encouraging. Probable mechanism of the tandem process of photocatalytic alcohol oxidation/Knoevenagel condensation promoted by Porph-UOZ-COF has been explored. I believe that this manuscript can be accepted for publication after some of the issues commented on below have been well addressed.
We thank the reviewer for his/her positive comments.
- The abbreviations of the monomers, DHPP and DBBA, should be provided at the first time its full name appears. The monomer abbreviations should also be labeled in Scheme 1. The synthesis of DHPP in Scheme 1 is not given in detail in this manuscript and this part in Scheme 1 is suggested to be deleted.
REPLY: As suggested by this reviewer, the abbreviations were provided in the abstract and the scheme 1 was thoroughly revised.
- In general, simulated X-ray powder diffraction pattern of the COF structure should be given for comparison, indicating that the obtained crystalline sample is the target product.
REPLY: We have cited carefully the relevant reference for comparison. However, we are not able to provide CIF crystalline data to extract their data for comparison.
- There are several types of reactive oxygen species, but only singlet oxygen is mentioned in the manuscript. Are there any other reactive oxygen species detected? The specific scavengers to reactive oxygen species should be added to the reaction process to observe the catalytic performance variation.
REPLY: Thanks a lot for this constructive comment. Accordingly, we have now addressed the reviewer’s comment (please see Table 4) and discussed more in the main text.
- Some works on COFs for photocatalysis should be cited and discussed, e.g., Molecules, 2022, 27(22), 8002.
REPLY: The suggested reference is now added to the main text as ref# 19 and discussed.
Reviewer 2 Report
The manuscript, entitled “A porphyrin-based covalent organic framework for metal-free visible-LED-light driven one-pot tandem benzylic C-H oxygenation/Knoevenagel condensation reaction” by Hermenegildo Garcia and co-workers, deals with the synthesis, characterization, and photocatalytic application of the porphyrin-based covalent organic framework. This manuscript is well-written and the results are nicely placed. This manuscript consists of two parts. Synthetic approaches to prepare the COF are not new, but are well described and characterization data seems well presented. On the other hand, the second part (application) was very much similar to the author’s previous articles (Ref 43, 46). The manuscript has potential, but several points must be cleared before being considered for publication in Nanomaterials.
1. Porphyrin-based architectures are not only used for organic synthesis but can use for other purposes such as water remediation (Inorg. Chem. Front. 2022, 9, 1270-1280), hydrogen production (Inorg. Chem. 2021, 60, 6, 3988-3995). Therefore, the introduction must be amended including this point.
2. Reference format must be compatible according to the journal’s criteria.
3. For the synthesis of Porph-UOZ-COF, the purity of this product was a big concern. The self-condensation of 1,4-phenylenediboronic acid also leads to the B3O3 framework (Science, 2005, 310, 1166-1170).
4. In the case of solid-state 11B NMR spectrum, two distinct signals arise probably from either the C2O2B ring or the B3O3 ring (Science, 2007, 316, 268-272.). Therefore, the formation of self-condensation B3O3 ring cannot be ruled out.
5. Line 220~221: “Notably, in additional controls using the COF monomers as catalyst replacing the active Porph-UOZ-COF, the product yield was negligible or no progress occurs under these conditions (Entries 7 and 8).” The authors need to elucidate this contrasting feature.
6. Previously, the authors reported that CsCu2I3@PCN-222(Fe)-3.5 h-60°C (15) required only 15 h to yield 96% for benzylidenemalononitrile from benzyl alcohol (ACS Appl. Mater. Interfaces 2022, 14, 36515-36526.). Then, what was the significance of this present study?
7. H2O was present during catalytic experiments. But, the authors claim that only singlet oxygen was the only reactive species. Other reactive species such as hydroxy radical or peroxy radical can have some effect on the reaction. A scavenger test must be required to confirm the mechanism (Int. J. Mol. Sci. 2022, 23, 13702).
8. A tabular form is required for the comparison of the effectiveness of this catalyst with other reported materials (ACS Sustainable Chem. Eng. 2022, 10, 5315-5322).
9. Is there any effect of the LED light intensity at a different wavelength on the reaction yield?
Author Response
The manuscript, entitled “A porphyrin-based covalent organic framework for metal-free visible-LED-light driven one-pot tandem benzylic C-H oxygenation/Knoevenagel condensation reaction” by Hermenegildo Garcia and co-workers, deals with the synthesis, characterization, and photocatalytic application of the porphyrin-based covalent organic framework. This manuscript is well-written and the results are nicely placed. This manuscript consists of two parts. Synthetic approaches to prepare the COF are not new, but are well described and characterization data seems well presented. On the other hand, the second part (application) was very much similar to the author’s previous articles (Ref 43, 46). The manuscript has potential, but several points must be cleared before being considered for publication in Nanomaterials.
We thank the reviewer for his/her positive comments.
- Porphyrin-based architectures are not only used for organic synthesis but can use for other purposes such as water remediation (Inorg. Chem. Front. 2022, 9, 1270-1280), hydrogen production (Inorg. Chem.2021, 60, 6, 3988-3995). Therefore, the introduction must be amended including this point.
REPLY: The suggested references are now added to the introduction as refs# 9 and 10.
- Reference format must be compatible according to.
REPLY: The reference format was now arranged according to the journal’s policy.
- For the synthesis of Porph-UOZ-COF, the purity of this product was a big concern. The self-condensation of 1,4-phenylenediboronic acid also leads to the B3O3 framework (Science, 2005, 310, 1166-1170).
REPLY: Thanks a lot for this constructive comment and your offer. Accordingly, we have revised the main text of solid-state 11B NMR. The suggested reference is also added to the revised manuscript as ref# 43.
- In the case of solid-state 11B NMR spectrum, two distinct signals arise probably from either the C2O2B ring or the B3O3 ring (Science, 2007, 316, 268-272.). Therefore, the formation of self-condensation B3O3 ring cannot be ruled out.
REPLY: Regarding to the previous reviewer’s comment, we have assigned the formation of B3O3 in the revised version. The suggested reference is also added to the main text as ref# 44.
- Line 220~221: “Notably, in additional controls using the COF monomers as catalyst replacing the active Porph-UOZ-COF, the product yield was negligible or no progress occurs under these conditions (Entries 7 and 8).” The authors need to elucidate this contrasting feature.
REPLY: As the reviewer suggested, the table and its entries are now better explained. The text in the revised version read as “Notably, in additional controls using the COF monomers and OMe derivative of DHPP as catalysts instead of an active Porph-UOZ-COF, the product yield was low or no progress occurs under these conditions (Entries 7-9). These consequences suggest that porphyrin groups in Porph-UOZ-COF act as photocatalytically active centers for this method. In addition, the higher product yield observed with Porph-UOZ-COF than that of the corresponding homogeneous catalyst DHPP (or OMe derivative of DHPP) is attributed to prevention of the well-known, efficiency diminishing, aggregation capability of porphyrin groups by construction of the COF. Furthermore, it can also related to the site isolation of porphyrin units in the framework.”
- Previously, the authors reported that CsCu2I3@PCN-222(Fe)-3.5 h-60°C (15) required only 15 h to yield 96% for benzylidenemalononitrile from benzyl alcohol (ACS Appl. Mater. Interfaces 2022, 14,36515-36526.). Then, what was the significance of this present study?
REPLY: Thank you for this useful comment. We have already mentioned the significance of this present study as given here “Also compared to the most efficient CsCu2I3@PCN-222(Fe) composite [ref 8], the present Porph-UOZ-COF catalyst enjoys a less elaborate synthesis to incorporate the mixed halide perovskite inside the MOF and a higher stability. In addition, Porph-UOZ-COF is a metal-free catalyst.”
- H2O was present during catalytic experiments. But, the authors claim that only singlet oxygen was the only reactive species. Other reactive species such as hydroxy radical or peroxy radical can have some effect on the reaction. A scavenger test must be required to confirm the mechanism (Int. J. Mol. Sci.2022, 23, 13702).
REPLY: According to your comment, we have performed quenching tests with various scavengers including sodium azide, 1,4-benzoquinone and isopropanol to define which active species were dominant in the photocatalytic process and the text was modified. Accordingly, 1,4-benzoquinone and isopropanol slightly decreased the production yield (Table 4, entries 2 and 3), demonstrating that superoxide radical anion (O2•−) and •OH were not substantial reactive species. But, the addition of sodium azide to the reaction exhibited significant inhibition, suggesting that singlet oxygen (1O2) was a dominant active species for this system (Table 4, entry 1). The suggested references are also added to the main text as ref# 58.
- A tabular form is required for the comparison of the effectiveness of this catalyst with other reported materials (ACS Sustainable Chem. Eng.2022, 10,5315-5322).
REPLY: A table summarizing the results from other reported materials including the reference was provided and discussed beyond what those of the previous discussions (please see new Table 3).
- Is there any effect of the LED light intensity at a different wavelength on the reaction yield?
REPLY: Thank you for your comment. It was also found that lower LED intensity can be employed as the light source but resulted in lower product yield (Table 1, entry 10).
Reviewer 3 Report
This manuscript by Prof. Oveisi, Prof. Garcia et al. describes the preparation of porphyrin-based covalent organic framework (COF) and the application toward photo-assisted alcohol oxidation/Knoevenagel condensation reaction. This manuscript may be suitable for the Journal. Some comments require to be addressed prior to the acceptance to the Journal.
(1) The porphyrin-COF is synthesized and characterized while the assignment of those 2 B-NMR peaks is required.
(2) The formation of a planar square-shape cavity shown in Scheme 1 is doubtful. Intercalating porphyrins to the formation of 3-D structural is possible. All possibilities should be examined. If the proposed structure is to be called, the author needs to provide additional information to support the structure.
(3) Knoevenagel condensation is a de-hydration process. Why is a mixture of MeCN/H2O used for the optimal synthesis condition? An explanation is needed.
(4) In Table 1, the OMe derivative of DHPP should be used as a catalyst alone for comparison.
(5) The EPR spectrum of the product from singlet oxygen oxidation of TEMP should be compared with that of pure TEMPO in the same condition.
(6) Evidences for the formation of aldehyde are needed.
(7) A list summarizing the results from Knoevenagel condensation is recommended. This list will include metal-free frameworks and representative metal-containing frameworks as well as the title compound, used as catalysts for the specific reaction.
Author Response
This manuscript by Prof. Oveisi, Prof. Garcia et al. describes the preparation of porphyrin-based covalent organic framework (COF) and the application toward photo-assisted alcohol oxidation/Knoevenagel condensation reaction. This manuscript may be suitable for the Journal. Some comments require to be addressed prior to the acceptance to the Journal.
We thank the reviewer for his/her positive and useful comments.
(1) The porphyrin-COF is synthesized and characterized while the assignment of those 2 B-NMR peaks is required.
REPLY: Thank you the review for the useful comment. Accordingly, we have assigned the peaks and revised the text regarding solid-state 11B NMR as follows “As shown in this Fig. 5, the COF shows two distinct signals arise probably from either the C2O2B ring (largely) or the B3O3 ring.” Also, new refs are now added to the main text.
(2) The formation of a planar square-shape cavity shown in Scheme 1 is doubtful. Intercalating porphyrins to the formation of 3-D structural is possible. All possibilities should be examined. If the proposed structure is to be called, the author needs to provide additional information to support the structure.
REPLY: We agree with the reviewer. As the reviewer suggested, the scheme 1 is now redrawn.
(3) Knoevenagel condensation is a de-hydration process. Why is a mixture of MeCN/H2O used for the optimal synthesis condition? An explanation is needed.
REPLY: Thanks for the nice comment. One of the essential challenges and eventual goals for organic reactions is to accomplish the reaction in water which is cheap, safe and leads to the development of environmentally benign processes. Research has shown that dehydration reactions in water have been proceeded with remarkable efficiency in the presence of the hydrophobic catalysts (Refs# 45-47; J. Am. Chem. Soc. 2012, 134, 1, 162-164; Green Chem., 2010, 12, 514-517; Catal. Commun., 2021, 154, 106304; Green Synth. Catal., 2020, 1(1), 79-82; Catal. Lett., 2022, https://doi.org/10.1007/s10562-022-04034-y). There are several behaviors in such catalysis: (1) providing hydrophobic pockets like enzymes, (2) condensing organic substrate therein, (3) intermediate stabilization, and (4) weak product binding. On the other hand, the use of aqueous medium as the reaction solvent makes the process environmentally friendly chemical process. Furthermore, the optimal conditions showed that the best reaction media was a mixture of CH3CN and H2O in which the reaction preceded smoothly, giving expected product in excellent yield within shorter reaction time. The most important references were now added to the main text.
(4) In Table 1, the OMe derivative of DHPP should be used as a catalyst alone for comparison.
REPLY: Thank you the review for the useful comment. According to your suggestion, we have performed the reaction in the presence of OMe derivative of DHPP and discussed its result in the main text (Please see Table 1, entry 10).
(5) The EPR spectrum of the product from singlet oxygen oxidation of TEMP should be compared with that of pure TEMPO in the same condition.
REPLY: EPR spin trapping technique with TEMP (2,2,6,6-tetramethylpiperidine) as spin trapper was used to determine the singlet oxygen photo-generation at room temperature. This is based on the detection of the TEMPO free radical (a strong 1:1:1 triplet signal) formed after oxidation of TEMP by singlet oxygen. This is supported by some references (refs#8 and 61) in the text.
(6) Evidences for the formation of aldehyde are needed.
REPLY: According to the review suggestion, GC/Mass spectrum of benzaldehyde as the intermediate was added to the supporting information (Fig. S4) to address this comment.
(7) A list summarizing the results from Knoevenagel condensation is recommended. This list will include metal-free frameworks and representative metal-containing frameworks as well as the title compound, used as catalysts for the specific reaction.
REPLY: As the reviewer suggested, a tabular form was provided for the comparison of the effectiveness of this catalyst with other reported materials as Table 3.
Reviewer 4 Report
"A porphyrin-based covalent organic framework for metal free visible-LED-light driven one-pot tandem benzylic C H oxygenation/Knoevenagel condensation reaction" is an interesting paper. However, I would have some comments about it.
1- In the title why not mentioning oxidation instead of CH oxygenation?
2- In Table 1, what is the LED wavelength and power?
3- In Table 1, if the solvent used is water, or aqueous solution, or organic/aqueous mix, is the reaction still working? Is benzoic acid detected?
4- In Table 2, only benzylic alcohols are oxidized, which is somehow easy because of the benzylic position of the oxidation. Is the method robust enough to work with linear alcohols, or hindered alcohols?
If answered, I recommend this paper for publication.
Author Response
A porphyrin-based covalent organic framework for metal free visible-LED-light driven one-pot tandem benzylic CH oxygenation/Knoevenagel condensation reaction" is an interesting paper. However, I would have some comments about it.
We thank the reviewer for his/her positive comments.
1- In the title why not mentioning oxidation instead of CH oxygenation?
REPLY: As the reviewer suggested, the title was changed.
2- In Table 1, what is the LED wavelength and power?
REPLY: The LED wavelength and power were already given in section 2.2.
3- In Table 1, if the solvent used is water, or aqueous solution, or organic/aqueous mix, is the reaction still working? Is benzoic acid detected?
REPLY: According to the reviewer suggestion, GC/Mass spectrum of benzaldehyde as the intermediate was added to the supporting information (Fig. S4) to address this comment. The reaction affords high yield of the target with a high selectivity and high conversion of benzyl alcohol (>99%).
4- In Table 2, only benzylic alcohols are oxidized, which is somehow easy because of the benzylic position of the oxidation. Is the method robust enough to work with linear alcohols, or hindered alcohols?
REPLY: Thank you for your comment. The catalyst was used with substrates from the other reported catalysts (Table 3) and showed a very high conversion of various benzyl alcohol derivatives with high selectivity towards the corresponding α,β-unsaturated nitriles under visible-LED-light. However, further work is under progress to extend this method to linear and hindered alcohols.
Round 2
Reviewer 2 Report
The manuscript has been sufficiently improved to warrant publication in Nanomaterials.
Reviewer 4 Report
The authors made the corrections and afforded the answers to the questions.
Thank you